# The Cascade Control of Natural Gas Pipeline Systems

**Kai Wen** [1,2], **Jing Gong** [1,*] **and Yan Wu** [3]

1  Beijing Key Laboratory of Urban Oil and Gas Distribution Technology,
   China University of Petroleum-Beijing, Beijing 102200, China; kewin1983@126.com
2  State Key Laboratory for Manufacturing Systems Engineering, Xi'an Jiaotong University, Xi'an 710049, China
3  PetroChina West East Pipeline Company, Shanghai 200122, China; xqdswy@petrochina.com.cn
*  Correspondence: ydgj@cup.edu.cn; Tel.: +86-135-0103-6944

**Abstract:** With the boost of natural gas consumption, an automatic gas pipeline scheduling method is required to replace the dispatchers in decision making. Since the state space model is the fundamental work of modern control theory, it is possible that the classical controller synthesis method can be used for the complicated gas pipeline controller design. In this paper, a cascade control algorithm is proposed based on the state space model that is used for the transient flow simulation of the natural gas pipelines. A linear quadratic regulator is designed following the classical optimal control theory. Finally, the transient process with different control methods shows the effectiveness of the cascade control using information of the entire pipeline. According to the hardware configuration of natural gas pipelines, automatic scheduling process is ready to deploy as one step to the intelligent natural gas pipelines.

**Keywords:** natural gas pipeline; cascade control; state space model; pipeline scheduling

## 1. Introduction

Most natural gas is typically transported by means of pipelines from the production and storage facilities to market regions. Supervisory Control And Data Acquisition (SCADA) system is a standard configuration in modern pipeline systems that works as the control center of natural gas pipelines, but rather focuses on the supervisory level [1]. Unlike the advances of the hardware configuration in pipelines, the scheduling scheme is still an empirical decision process [2]. Current industrial practice for natural gas pipeline control involves regulatory control loops along with manual intervention by pipeline dispatchers. The transient process is regulated by the real-time change of compressor speed and valves openness [3]. It includes inlet pressure regulation, outlet pressure regulation and flow regulation. When more than two parameters need to be adjusted at the same time, the PID controller choose the actuator according to the principle of "small value selection". With the PID controller, the stability control of local parameters is determined by the signal of inlet pressure, outlet pressure or flow deviation. The input, output and reference signals of the system are all local variables in the station. Since the control process is not affected by the entire pipeline state, it is called the local PID control.

During the scheduling process, the dispatchers adjust the parameters according to the state of the entire pipelines. The transient process is fast and steady under the operation of experienced dispatchers. Following the concept of unmanned vehicles, this paper attempts to develop a control algorithm replacing the experienced dispatchers, i.e., the SCADA system can schedule the piping systems independently without the manual intervention. The controller design method employed may not be the frontier theory. The way to solve the problem is inspiring with the rise of Artificial Intelligence (AI) technology.

The hierarchical control structure of natural gas pipelines is shown in Figure 1. According to users' demand and pipeline capacity, the pipeline scheduling center carries out gas supply balance analysis,

and formulates the medium and long term gas supply plan and pipeline scheduling scheme [4,5]. The mid-long term and short term optimization process is calculated off-line and the result is the guideline of one month of operation. The short term optimal operation is to get the steady flow and pressure along the pipelines [6]. Since the gap between two steady states is large, it is impossible to achieve by just one set point command to the PID controllers along the pipelines [7]. The state of the piping systems has to move gradually by the set point command sequences based on the dispatchers' experience [8]. They make decisions usually following system dynamic characters with the state of the entire pipelines.

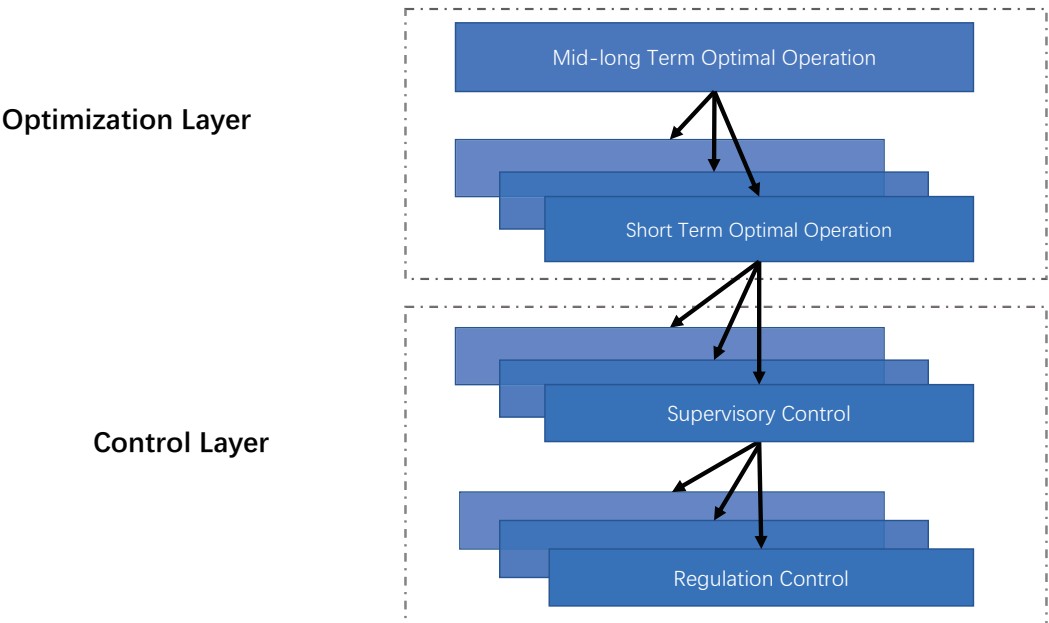

**Figure 1.** The hierarchical structure of the natural gas pipeline control system.

On the other hand, the transient process control is more difficult mathematically than the steady state because of the fluctuations in gas supply and gas consumption [9]. The dynamic behaviors of natural gas pipelines are characterized by the fluid equations consisting of a set of nonlinear hyperbolic partial differential equations [10]. A control oriented mathematical model and the corresponding algorithm are necessary for the automatic scheduling process. Until recently, the state space model is used for the transient flow simulation in gas pipelines [11]. Since the state space model is the fundamental work of modern control theory, it is possible that the classical controller synthesis method can be used for the complicated gas pipeline controller design [12].

In this paper, a cascade control algorithm is deployed in the SCADA system as an additional control level between the scheduling process and the PID controllers. The virtual controller gives the continuous set point command sequences based on two successive steady hydraulic states in real time [13]. By using the information from the pipeline parameter measurements, the continuous set point sequences are sent to the local PID controllers to improve the performance of an entire natural gas pipeline system. By regulating the speed of the compressors and the openness of the valves following the upper layer command, the local PID controllers adjust the process parameters with not only the local information but also the entire system information. Thus, the transient process of the entire system is controlled by the algorithm automatically instead of the dispatcher.

The rest of the paper is organized as follows. In Section 2, the hierarchical structure of the cascade control is explained in detail. Based on the state space model, the cascade control algorithm is implemented with a linear quadratic indicator that is described. In Section 3, an example is used to analyze and simulate the transient behaviors of a gas pipeline. The scheduling process under different schemes are compared to show the effectiveness of the cascade control algorithm.

## 2. Cascade Control Algorithm

In cascade control, the entire pipeline is divided into different subsystems according to the time scale of pipe sections and the compressors along the pipeline. The outer loop of cascade control is coordinating the behavior of the entire pipeline system based on the linear dynamic model of the subsystems. The output of the outer loop is taken as the reference values of the inner subsystems. The disturbances and expected values of the outer loop is affected by gas market and end users. While the inner loop is related to the disturbance of the process parameters which is affected by the adjustment of the compressors and valves.The inner loop of cascade control is the local PID controllers for the set point value. The inner loop handles the rapid disturbances of local units and keeps the process parameters following the settings given by the outer loop as soon as possible, as shown in Figure 2.

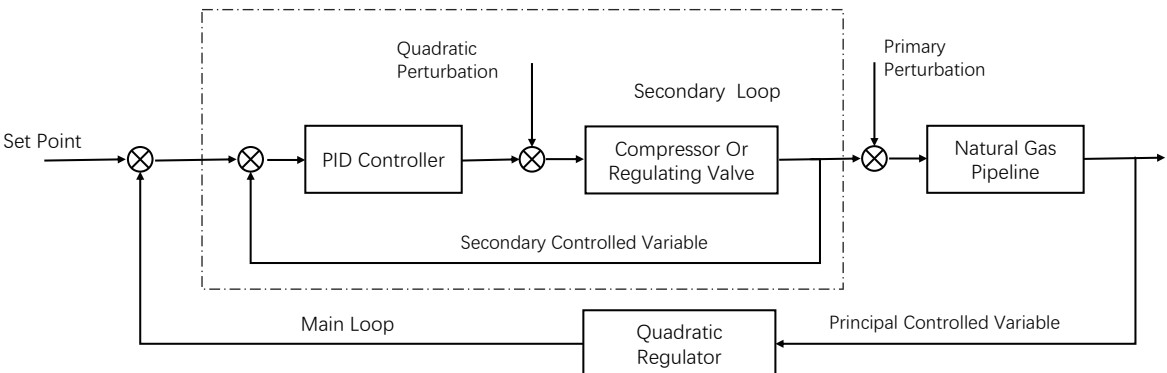

**Figure 2.** The cascade control system of the natural gas pipeline.

### 2.1. State Space Model

To get the cascade control algorithm, a mathematical model of the dynamic properties of the gas pipelines has to be established for the outer loop calculation. The modeling process starts from the fundamental governing equations of fluid.

$$\frac{\partial \rho}{\partial t} + \frac{\partial (\rho \mathrm{u})}{\partial x} = 0 \tag{1}$$

$$\frac{\partial p}{\partial x} = -\frac{\partial (\rho \mathrm{u})}{\partial t} - \lambda \frac{u|u|}{2D}\rho - \rho g \sin\alpha \tag{2}$$

$$\frac{\partial Q}{\partial x}(\rho \mathrm{u}) = \frac{\partial}{\partial t}[(\rho)(\hat{u} + \frac{\mathrm{u}^2}{2} + g\Delta h)] + \frac{\partial}{\partial x}[(\rho)(H + \frac{\mathrm{u}^2}{2} + gs)] \tag{3}$$

$$\frac{p}{\rho} = ZRT, \tag{4}$$

where $\rho$ is the gas density, $kg/m^3$; $D$ is the pipeline inner diameter, $m$; $g$ is the gravitational acceleration, $m/s^2$; $\alpha$ is the inclination; $T$ is the gas temperature, *Kelvins* (K); $R$ is the gas constant, $kJ/(kg \cdot K)$; $\lambda$ is the friction factor of the pipeline; $Z$ is the gas compressibility factor; $u$ is the gas axial velocity, $m/s$; $\frac{\partial Q}{\partial x}$ is the heat flow into the pipe per unit length of pipe and per unit time, $J/(m \cdot s)$; $\hat{u}$ is the specific internal energy, $J/kg$; $\Delta h$ is the elevation, $m$; $H$ is the specific enthalpy, $J/kg$; $x$ is pipeline coordinate, $m$; and $t$ is time, $s$.

The method of characteristics, explicit and implicit finite difference methods, and finite element methods have been used to solve the governing equations to obtain good simulation results. However, a control oriented mathematical model is necessary for the automated scheduling process. The modeling process consists of the following steps:

- The equivalent transfer functions are derived from the governing Equations (1)–(4).

- The series form of the models in time domain is obtained using the convolution theorem.
- A few dominant flow eigenmodes are used to construct an efficient reduced-order model, i.e., ordinary differential equation (ODE) model.

As seen from the above, the state space models are to neglect some of the terms in partial differential equations (PDE) and maintain an appropriate level of accuracy at the same time. By the convenience of the state space model, it can be easily applied for the cascade controller design.

Following the derivation process in [14], the state space models (Equation (5)) is ready to be obtained, where the gas pressure at inlet ($P_{in}$) and the mass flow rate at the outlet ($M_{out}$) are the input vectors, and the pressure at the outlet ($P_{out}$) and the gas mass flow rate at the inlet ($M_{in}$) are the output vectors.

$$
\begin{aligned}
\dot{x} &= \mathbb{A}x + \mathbb{B}u \\
y &= \mathbb{C}x + \mathbb{D}u
\end{aligned}
\tag{5}
$$

where $u = [P_{in}, M_{out}]$ and $y = [P_{out}, M_{in}]$

$$
\mathbb{A} = \begin{bmatrix}
-\frac{a_1}{a_2} & 1 & 0 & 0 \\
0 & -\frac{1}{a_2} & 0 & 0 \\
0 & 0 & -\frac{a_1}{a_2} & 1 \\
0 & 0 & -\frac{1}{a_2} & 0
\end{bmatrix}, \quad
\mathbb{B} = \begin{bmatrix}
0 & -\frac{k_2 T_{21}}{a_2} \\
\frac{k_1}{a_2} & -\frac{k_2}{a_2} \\
\frac{T_{11}}{a_2} & 0 \\
0 & \frac{1}{a_2}
\end{bmatrix}, \quad
\mathbb{C} = \begin{bmatrix}
1 & 0 & 0 & 0 \\
0 & 0 & 1 & 0
\end{bmatrix}.
\tag{6}
$$

Since the state equation of pipe section is multi-dimensional, the state dimension is related to the number of pipe sections. There are two input variables and the output variables are related to the measurement points. Therefore, $x$, $u$, and $y$ in the following formula are multi-dimensional vectors and the system is a multiple-input multiple-output (MIMO) system.

$$
G \begin{cases}
\dot{x}(t) = Ax(t) + Bu(t) \\
y(t) = Cx(t)
\end{cases}
\tag{7}
$$

## 2.2. Linear Quadratic Regulator

The natural gas pipelines are self-balanced systems. It is easy to get a PID cascade control by tuning the parameters of the inner and outer loop PID controllers. To get the desired steady state quickly and cost effectively, the linear quadratic regulator (LQR) is used for the controller design.

$$
J = \frac{1}{2} \int_0^\infty (x^T Q x + u^T R u) dt,
\tag{8}
$$

where $J$ is the optimization of performance index. $Q$ is a positive semi define symmetric weight matrix, and $R$ is a positive define symmetric weight matrix. In the LQR control, different $Q$ and $R$ are selected for different performance requirements. $Q$ is related with the regulation speed and $R$ is related with the control ability. The optimization of $J$ is to minimize the transient time and energy consumption. An experienced dispatcher of the natural gas pipeline can adjust the pipeline to a new steady state in very short time. Thus, both the transfer time and the cost are meaningful to the cascade control when the accuracy of the end state is guaranteed by the LQR controller. In actual pipeline system, all states of the dynamic equation of the pipeline section is hard to obtain. Thus, the controller design is based on the measured pressure and flow rate. The output feedback LQR is solved in time domain. The value of initial state $x(0)$ will affect the calculation of the optimal performance. To avoid this problem, the optimization of performance index $J$ is turned into searching the minimum value of expected value $E(J)$.

$$
E(J) = \frac{1}{2} E\{ \int_0^\infty (x^T Q x + u^T R u) dt \} = \frac{1}{2} E\{ x^T(0) P x(0) \} = \frac{1}{2} tr(PX)
\tag{9}
$$

$$0 = A_c^T P + A_c P + C^T \mathbf{K}^T R \mathbf{K} C + Q \tag{10}$$

$$0 = A_c S + S A_c^T + X \tag{11}$$

$$\mathbf{K} = R^{-1} B^T P S C^T (C S C^T)^{-1}, \tag{12}$$

where $X = E\{x(0)x^T(0)\}$ is an $n \times n$ autocorrelation matrix of initial state and it is assumed as identity matrix $I$ in the calculation. $A_c = A - B\mathbf{K}C$ is asymptotically stable, $S$ is the Lagrange multiplier, and $P$ and $S$ are solutions of the Lyapunov equation. An iterative calculation of feedback gain matrix $\mathbf{K}$ is as follows.

- (1) Initialization:
  Let $k = 0$,
  the initial $\mathbf{K}_0$ is solved by $A - B\mathbf{K}_0 C$ asymptotically stable.
- (2) The $k$ th iteration:
  Let $A_k = A - B\mathbf{K}_k C$,
  $P_k$, $S_k$ are solved following Lyapunov equation.
- (3) Solving iterative direction of gain matrix $\Delta \mathbf{K}$:
  Let $J_k = \frac{1}{2} tr(P_k X)$,
  $\Delta \mathbf{K} = R^{-1} B^T P_k S_k C^T (C S_k C^T)^{-1} - \mathbf{K}_k$.
- (4) Solving iteration matrix:
  Let $\mathbf{K}_{k+1} = \mathbf{K}_k + \alpha \Delta \mathbf{K}$ and $\alpha \in (0, 1]$ satisfies the following conditions,
  $A - B\mathbf{K}_{k+1} C$ is asymptotically stable,
  and $J_{k+1} = \frac{1}{2} tr(P_{k+1} X) \le J_k$.
- (5) Determine whether iteration is needed:
  If the difference between $J_{k+1}$ and $J_k$ is small enough,
  then go to (6);
  otherwise,
  let $k = k + 1$ and go to (2).
- (6) Termination:
  Let $\mathbf{K} = \mathbf{K}_{k+1}$,
  $J = J_{k+1}$.

Here, the output feedback gain matrix can be found by the iterative algorithm if it satisfies the following four conditions.

- There is a matrix $\mathbf{K}$ to make $A_c = A - B\mathbf{K}C$ asymptotically stable. Since the pipeline system is a self-balance system, the initial matrix can be calculated by using the eigenvalue assignment method.
- The system output matrix $C$ is full row rank.
- To get all control inputs in the performance index, the weighted matrix $R$ must be positive definite.
- The state weight matrix $Q$ is semi positive definite. $(\sqrt{Q}, A)$ is detectable, which means that the unstable state in the index is weighted.

## 3. Case Studies of the Cascade Control of Natural Gas Pipelines

A natural gas pipeline with a total length of 169 km has only the initial station and the terminal station. The initial station has a speed regulating compressor that is adjusted by the outlet pressure. The terminal station has a regulating valve that is adjusted by the outlet pressure. To acquire the exact behavior of the pipeline, the Stoner Pipeline Simulator (SPS) software is used to simulate the pipeline. SPS is a powerful commercial software package and capable of analyzing and predicting the hydraulic performance of both liquid and gas pipeline systems accurately. SPS has the ability to simulate the operating characteristics of almost all configurations of pipelines and equipments as they are subjected to various control strategies and operating scenarios.

The simulation model in SPS is shown in Figure 3. The PID parameters of the local controllers are listed in Table 1. The initial working condition is a steady state that the pressure and flow rate along the pipeline are shown in Figure 4. Two step downs of the flow rate means that there are two distribution points along the pipeline.

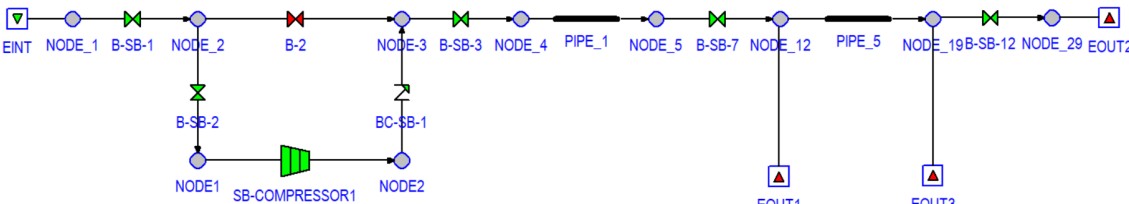

**Figure 3.** Stoner Pipeline Simulator (SPS) model of the pipeline.

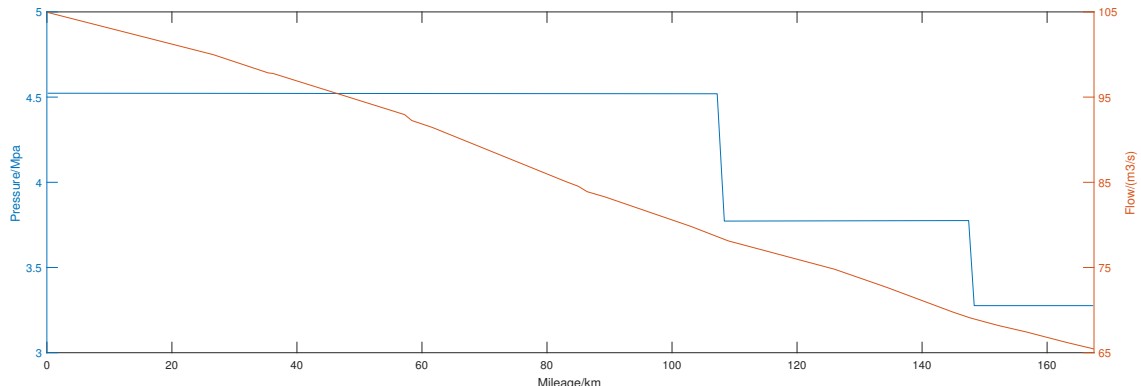

**Figure 4.** Steady state diagram.

**Table 1.** Parameters of PID controller.

| Station | Controller Type | Proportional Gain $K_c$ | Integral Time $T_I$/$min$ | Differential Time $T_D$/$min$ |
|---|---|---|---|---|
| First station | Pressure Controller | 1.12 | 0.2 | 0 |
| terminal station | Pressure Controller | 0.5 | 0.2 | 0 |

When there is a disturbance of the flow rate at the distribution station, the compressor in the initial station is able to adjust itself accordingly. In systems with only local PID controllers, the pressure and flow rate at each station are set and adjusted manually by the dispatchers. It means that the systems is not fully automatically controlled but scheduled by human beings. The dispatchers have to watch the working conditions of the pipelines and perform the regulating operations all the time in reality.

The pressure along the pipeline is chosen as the main variable to be controlled. The pressure at the compressor outlet is taken as the secondary variable. The terminal control valve keeps the outlet pressure of the control valve at a constant value 5 MPa. In design of the cascade control system, *Q* is a diagonal matrix and its larger elements means the importance of the corresponding state variables. *R* is also a diagonal matrix and its larger elements means the importance of the corresponding control variables. *Q* and *R* could be set as different values under different transient process requirement. In this case, matrices *Q* and *R* are set as the identity matrix, where the performance index *J* is to balance the transient time and control energy cost.

The optimal control model is solved by steps given in Section 2 and the feedback matrix **K** is obtained as follows:

$$\mathbf{K} = [0.0691 \quad -0.0563 \quad 0.6656 \quad 0.0559 \quad -0.5525 \quad 0.9497 \quad -0.6233]. \tag{13}$$

The set points of the compressor PID controller are not given by dispatchers manually, but calculated by Equation (14) via the SCADA system.

$$u = -\mathbf{K}y. \tag{14}$$

To verify the effectiveness of the algorithm and to eliminate the contingency, six initial states were simulated. The outlet pressure of the compressors and the terminal flow rate were considered as the disturbance of the system. With the cascade control, the outlet pressure of the control valve was stabilized. The relative error of the final outlet pressure of the compressor was less than 5%. The results are given in Table 2. These results show that the cascade control could keep end users with stable pressure whenever there was a disturbance at different points of the system.

**Table 2.** The validation results of controller.

| | Process Variable | Disturbance | Final Result/MPa | Absolute Error/Mpa | Relative Error |
|---|---|---|---|---|---|
| 1 | Outlet Pressure | 5.2 MPa | 4.9991 | −0.0009 | 0.45% |
| 2 | Outlet Pressure | 4.8 MPa | 5.0006 | 0.0006 | 0.30% |
| 3 | Outlet Pressure | 6.0 MPa | 4.9940 | −0.0060 | 0.60% |
| 4 | Outlet Pressure | 4.0 MPa | 5.0467 | 0.0467 | 4.67% |
| 5 | Terminal flow | 75 m$^3$/s | 5.0010 | 0.0010 | 3.23% |
| 6 | Terminal flow | 65 m$^3$/s | 4.9987 | −0.0013 | 3.51% |

Currently, the goal of the scheduling given by the dispatchers is to stabilize the outlet pressure connected with end users. The scheduling of the pipeline working conditions are to satisfy the need of the users in the mid-long term. However, the dispatchers will not able to adjust the pipeline following the wave of the consumption of end users. A better operation is to make the actual flow rate of the terminal station vary according to the users' requirement. Here, it was assumed that the flow rate changes during 24 h, as shown in Figure 5. The pipeline was in transient state because of the continuous change of flow rate. The dynamic control of natural gas pipeline was to let the flow rate of the natural gas in the terminal end following the need the of the users. The criterion is that the transient process is stable and fast.

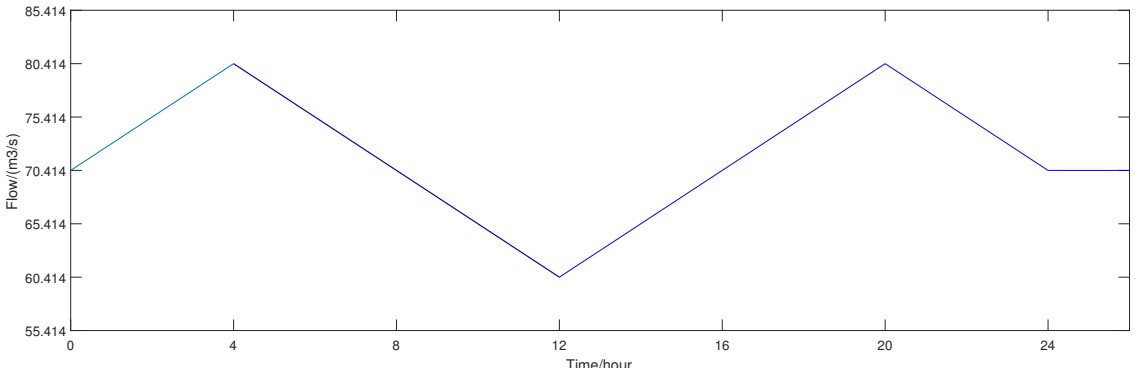

**Figure 5.** Flow rate changes of natural gas pipeline terminal station.

With only the PID controllers and the cascade control algorithm, the output of the compressor is shown in Figure 6. The transient process of the entire line was 38.13 h for only local PID controllers with the help of an dispatcher. The adjustment of the compressor's speed was not smooth and steady. The dispatcher might need to improve the scheduling process. However, it would require more practice and experience. The time under cascade control was 29.55 h. The adjustment process of the compressor was perfect. Therefore, the cascade control is an effective control algorithm for the pipeline in transient situations.

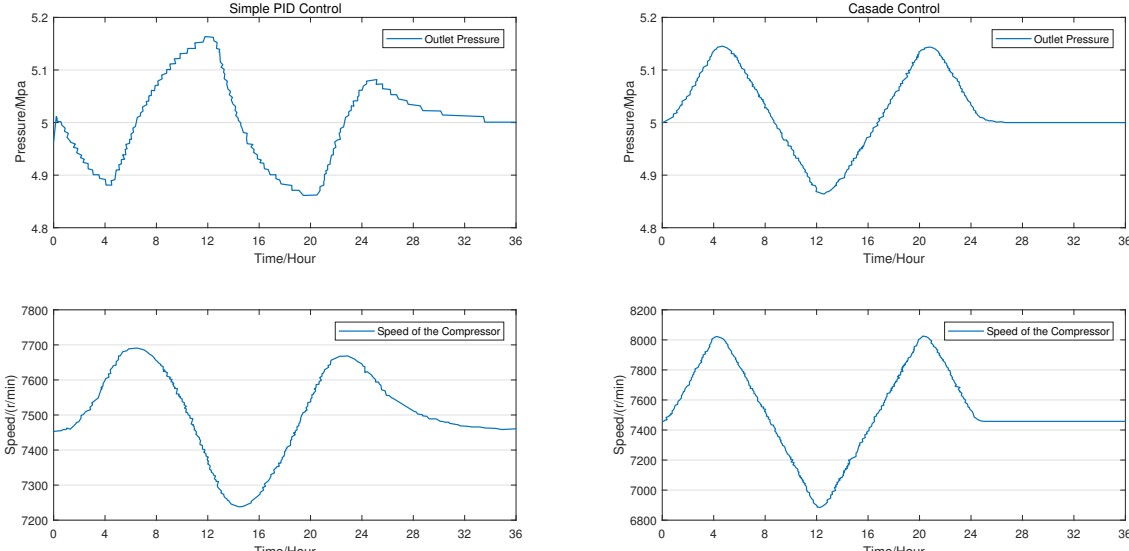

**Figure 6.** Comparison of compressor speed and outlet pressure under different control methods.

## 4. Conclusions

In this paper, a cascade control algorithm of natural gas pipelines is established as primary attempts to the automatic scheduling of pipelines. To make full use of the existing instruments, the inner loop of the cascade control uses the PID controllers. The main loop adopts the linear quadratic regulator of the entire pipeline system. System performances with dispatchers and the cascade control given commands automatically are compared and analyzed. The effectiveness of the cascade control is proved. The cascade control algorithm shows that the use of information of entire systems can make the transient process better than the traditional control method. Based on gas consumption of the end users, the pipeline is able to guide itself to the right working conditions fast and accurately.

As a small step of the improvement of the pipeline technique, the commonly used cascade control algorithm is an attempt to make the pipeline schedule automatically. The cascade control itself is a classical method in control theory. The algorithm implemented in the SCADA system connects the instruments along the pipelines and the actuators such as compressors and regulation valves in the filed. The closed loop control of the scheduling process makes full use of the hardware and software configurations of the pipelines. The development of AI science is a good opportunity for advances in natural gas pipelines. The small step to the intelligent system can release the dispatchers from the full time scheduling operations of the natural gas pipelines.

**Author Contributions:** This review article was jointly written and proof-read by all authors. K.W. gave the modeling, drafted the outline and structure as well as conclusion. J.G. brought out the concept and designed the study. Y.W. collected references and analyzed the situation in the field.

**Funding:** This research was funded by the National Natural Science Foundation of China (No. 51504271), the National Science and Technology Major Project of China (2016ZX05028004-001), and the National Key R&D Proram of China (2017YFC0805800).

**Acknowledgments:** This work was supported by the State Key Laboratory for Manufacturing Systems Engineering, Xi'an Jiaotong University.

**Conflicts of Interest:** The authors declare no conflict of interest.

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
