# Peer review of "The Cascade Control of Natural Gas Pipeline Systems"

_applsci, doi:10.3390/app9030481_

Round 1
Reviewer 1 Report
1. The authors mention that the model for the system under study consists of nonlinear hyperbolic partial differential equations. and they are using LQR as their control strategy which is for linear system. The authors didn't explain how did they analyse their system as linear system?
How did they get the model explained in equation 5 and 6.
2. What GA algorithm was used for optimization?
3. The quality of the figures is not good.
4. Explanation in section 4 is very weak and vague.
5. Results and conclusion section is weak. There was no explanation of any figure.
Overall, for the control system experts this paper in its present state offers nothing new. However, it may attract their attention because of the system. However, the authors needs to explain the system in a much better way.
Author Response
Dear Reviewer,
Thank you very much for your time and great efforts on our manuscript. Your comments are all valuable and very helpful for revising and improving our paper, as well as the important guiding significance to our researches. In accordance with your kind suggestions and comments, the manuscript has been revised comprehensively. The paper is reorganized more reasonably and the study process is explained in a better way. The responses to the questions are list as following.
COMMENT 1: The authors mention that the model for the system under study consists of nonlinear hyperbolic partial differential equations. and they are using LQR as their control strategy which is for linear system. The authors didn't explain how did they analyse their system as linear system? How did they get the model explained in equation 5 and 6.
RESPONSE:The modeling methods are with the following steps:
1.The equivalent transfer functions are derived from the governing equations.
2. The series form of the models in the time domain is obtained using the convolution theorem.
3. A few dominant flow eigenmodes are used to construct an efficient reduced-order model, i.e. ODE(ordinary differential equation) model.
The detailed process of deriving the state space models are given in our last paper (i.e. Wen, K.; Xia, Z.; Yu, W.; et al. 2018. A new lumped parameter model for natural gas pipelines in state space. Energies 11(8):1-17.). To make the paper to the point, only the citation is given in line 101 and the model is directly listed as equation 5 and 6.
COMMENT 2: What GA algorithm was used for optimization?
RESPONSE:After the discussion with other authors, we agree that the GA algorithm is a redundancy in this paper. The paper aims to give an active control method that has not yet implemented by the current system. The optimization of the control algorithm is another problem which will lead the study to a promiscuous situation. Thus, the GA optimization is deleted in the manuscript.
COMMENT 3: The quality of the figures is not good.
RESPONSE:Thank you very much for your kindly reminder. The manuscript is written by Latex. And there is something wrong with my GSview. We are trying to find out how to solve it. After all, the jpg format figures will be uploaded for the final print version.
COMMENT 4: Explanation in section 4 is very weak and vague.
RESPONSE:Sorry for the insufficient discussion. The comparison part is reorganized this time. Without the GA optimized cascade control, the advantages of the cascade control to the operators with the local controller are ready to find. The effectiveness of the cascade control is explained in detail in this revision.
COMMENT 5: Results and conclusion section is weak. There was no explanation of any figure.
RESPONSE:The final part is rewritten. And the explanation of the figures is added accordingly. The original version of the paper is confusing. The revision of the paper is refined comprehensively both the introduction part and the conclusion part to make the point more clear.
Reviewer 2 Report
The paper is interesting and appropriate for the Journal.
However, I believe there are parts that can bo omitted (for instance, the part reporting the control equations, which can be found in any students' manual), and others that should be better clarified.
In particular, the integration algorithm should be stable (which one has been used?), otherwise recorded instabilities could be numerical and not physical Anyhow I assume the Authors used a commercial model, which are normally stable (even too much, sometimes).
I understood the parameter to be minimized is J, but it is not clear its definition/physical meaning.
Say "a GA model has been used" is not enough: there are dozens of these algorithms. Which one has been used? Individuals I think they are real numbers (see Table 3): how can the Authors be sure the whole space has been examinated (see the differences among the use of real and binary). Compared with the number of parameters to be calibrated, how many individuals have been used?
Figure 14 is not clear to me: it seems the best fitness has been reached during the first run (population) and then never improved. So... the GA do not work? Why the mean fitness do not decrease during generations?
I think these points should be clarified before publishing.
Author Response
Dear Reviewer,
Thank you very much for your time and great efforts on our manuscript. Your comments are all valuable and very helpful for revising and improving our paper, as well as the important guiding significance to our researches. In accordance with your kind suggestions and comments, the manuscript has been revised comprehensively. The paper is reorganized more reasonably and the study process is explained in a better way. The responses to the questions are list as following.
COMMENT 1: there are parts that can bo omitted (for instance, the part reporting the control equations, which can be found in any students' manual), and others that should be better clarified.
RESPONSE:Thank you very much for your suggestion. Several paragraphs in Section 1 are deleted and Section 1 is reorganized to make the problem and situation more clear. The reason of using cascade control is explained in Section 1.
COMMENT 2: In particular, the integration algorithm should be stable (which one has been used?), otherwise recorded instabilities could be numerical and not physical Anyhow I assume the Authors used a commercial model, which are normally stable (even too much, sometimes).
RESPONSE:Yes, the commercial software SPS and the build-in model is used to simulation the transient behavior of the natural gas pipeline. The cascade control algorithm is written in MATLAB environment. These two software communicate with each other via soft OPC interface. The commercial software is more accurate and easy to get stable results.
COMMENT 3: I understood the parameter to be minimized is J, but it is not clear its definition/physical meaning.
RESPONSE:The explanations of parameter J is add in Line 108-112 as following.
“In the LQR control, different Q and R are selected for different performance requirements. Q is related with the regulation speed and R is related with the control ability. For an experienced operator of the natural gas pipeline, he is able to adjust the pipeline to a new steady state in very short time. Thus, both the transfer time and the cost are meaningful to the cascade control when the accuracy of the end state is guaranteed by the LQR controller.”
COMMENT 4: Say "a GA model has been used" is not enough: there are dozens of these algorithms. Which one has been used? Individuals I think they are real numbers (see Table 3): how can the Authors be sure the whole space has been examinated (see the differences among the use of real and binary). Compared with the number of parameters to be calibrated, how many individuals have been used?
Figure 14 is not clear to me: it seems the best fitness has been reached during the first run (population) and then never improved. So... the GA do not work? Why the mean fitness do not decrease during generations?
RESPONSE:After the discussion with other authors, we agree that the GA algorithm is a redundancy in this paper. The paper aims to give an active control method that has not yet implemented by the current system. The optimization of the control algorithm is another problem which will lead the study to a promiscuous situation. Thus, the GA optimization is deleted in the manuscript.
Round 2
Reviewer 2 Report
I believe the paper has improved, going straight to the point.
I am still wondering whether the objective function has a physical meaning or not (and in case it has, which is).
I am not a native speaker, but sometimes the used English made me unconfortable.
I am wondering whether the scale of figure 6, pressure, PID control (Pressure [MPa]) is correct. It seems that with this control, PID, variations are practically zero...
Author Response
Dear Reviewer,
Thank you very much for your time and great patience. Your comments are all valuable and very helpful for revising and improving our paper. First of all, the manuscript is polished by an English native speaker. The responses to the questions are list as following.
COMMENT 1: Whether the objective function has a physical meaning or not (and in case it has, which is).
RESPONSE: “In design of the cascade control system, matrices $Q$ and $R$ are set as the identity matrix, where the performance index $J$ is to balance the transient time and control energy cost. Here, $Q$ and $R$ could be set as different values under different transient process requirement. The identity matrix is an option to balance the time and energy cost. ” The explanation is add in Line 174-177. And the objective function does not have a specific physical meaning but is used to illustrate the optimization process.
COMMENT 2: Whether the scale of figure 6, pressure, PID control (Pressure [MPa]) is correct. It seems that with this control, PID, variations are practically zero.
RESPONSE:We double check the input and output signals and the calculation process. The pressure and speed values under PID control and cascade control are as in Figure 6. The requirement of end users is set to 70 m^3/s after a 24 hour period, which may mislead reviewers. We have extended the line to 26 hours to make it clear.
